# Characteristics of RNA Stabilizer RNApro for Peripheral Blood Collection

**DOI:** 10.3390/diagnostics14100971

**Published:** 2024-05-07

**Authors:** Stefano Gambarino, Ilaria Galliano, Anna Clemente, Cristina Calvi, Paola Montanari, Anna Pau, Maddalena Dini, Massimiliano Bergallo

**Affiliations:** 1Department of Public Health and Pediatric Sciences, Immunopathology Laboratory, Medical School, University of Turin, Piazza Polonia, 94, 10126 Turin, Italy; gambarino.stefano@gmail.com (S.G.); ilaria.galliano@unito.it (I.G.); anna.clemente@unito.it (A.C.); cristina.calvi@unito.it (C.C.); paola.montanari@unito.it (P.M.); anna.pau@unito.it (A.P.); maddalena.dini@unito.it (M.D.); 2BioMole srl, Via Quarello 15/A, Turin, 10135, Italy

**Keywords:** RNA stabilizers, RNA extraction, RNApro, guanidinium thiocyanate, GAPDH

## Abstract

Peripheral blood is the most practical tissue for human immune system gene expression profiling because it is easily accessible, whereas the site of primary infection in certain diseases may not be easily accessible. Due to the ex vivo instability of RNA transcripts, a key challenge in the gene expression analysis of blood samples is the rapid sample handling and stabilization of the mRNA by adding an RNA preservative (PAXgene^TM^ Blood RNA Tubes, Tempus^TM^ Blood RNA tubes, RNAlater Stabilization Reagent, RNAgard^®^ Blood Tubes). BioMole (Turin, Italy) has developed a novel blood stabilizer, called RNApro, in which RNA is stabilized during phlebotomy and sample storage. In this study, RNApro performance intended as RNA yield, integrity, and stability was evaluated. Our results show that blood samples stored at −80 °C and re-extracted after 7 years show no differences in terms of quantity, quality, and amplificability. The samples in the RNAlater stabilization solution can be stored at room temperature for up to one week or at 4 °C for up to one month. Similar results can also be observed for PAXgene tubes, Tempus tubes, and RNAgard tubes. In agreement with these data, the RNApro stabilization solution preserves the RNA from degradation for up to 1 month at 4 °C and 1 week at room temperature. RNApro can be stored indifferently at −80, −20, 4 °C, or room temperature for up to 2 months after, and then could be stored at −80 °C for up to seven years. In summary, our study is the first to analyze the performance of an RNA stabilizer called RNApro. We can conclude that several studies have shown significant differences in gene expression analysis when the sample was preserved in different RNA stabilizers. Therefore, it is desirable to standardize the method of nucleic acid conservation when comparing data from transcriptomic analyses.

## 1. Introduction

Peripheral blood (liquid biopsy) is routinely collected by biobanks for biomarker analysis for scientific research projects, epidemiological studies, and medical applications. The major limitation associated with RNA is the phenomenon of RNA degradation during blood collection and storage. Both the sample collection and the RNA purification procedures influence the data outcome [1]. Comparing DNA and RNA, the latter is more susceptible to non-enzymatic degradation due to the presence of a 20-hydroxyl group. The 20-hydroxyl group has the potential to cleave the backbone by attacking the neighboring phosphodiester bond. In addition, single-stranded nucleic acids are less stable than double-stranded nucleic acids. The stabilization of RNA against chemical and enzymatic degradation leads to improved biological function. RNA can be both chemically degraded when certain chemical entities are present and physically unstable when higher-order structures change. Physical instability includes the loss of kinetically stable structural conformation (secondary or tertiary structure), aggregation, and precipitation [2]. The chemical degradation of RNA in solution can occur via hydrolysis and/or oxidation [3]. Hydrolysis occurs mainly via RNA strand breakage at the P-O5′ ester bond due to a nucleophilic attack of the 2′OH group of the ribose at the phosphodiester bond via a transesterification reaction. This process is catalyzed enzymatically by nucleases or non-enzymatically through water itself and factors such as Brønsted acids and bases or divalent cations [4]. Regardless of the structural conformation (single-stranded or double-stranded), the acidic or alkaline hydrolysis of phosphodiester bonds can occur anywhere in the sequence. The oxidation of nucleotides occurs mainly through reactions with reactive oxygen species (ROS). Although oxidation is a likely cause of RNA degradation, hydrolysis is considered the major mechanism for RNA degradation. Chemical degradation plays a greater role in RNA degradation and in reducing its biological activity than physical instability. This is all the more true the greater the number of nucleotides there are. There are many factors that influence the stability of unencapsulated or naked RNA in solution. RNA length, polyA tail and 5′-cap integrity, excipients (such as buffer species, salt, etc.), solution pH, and enzymes such as nuclease and divalent cations such as Mg^2+^ or Ca^2+^ are some of the factors that influence RNA stability in solution. RNA length is negatively correlated with RNA stability. With increases in the length of RNA, decreases in its stability are observed. This might be simply due to an increase in the putative sites of cleavage with an increase in the length of RNA. Transcriptional profiling from whole blood has become a powerful tool for biomarker discovery in clinical medicine and health research [5]. This technology offers the possibility of studying the altered expression of genes using quantitative polymerase chain reaction (RT-qPCR), RNA sequencing, or microarrays and can lead to a better understanding of the regulatory processes in health and disease processes. Peripheral blood is the most practical tissue for human immune system gene expression profiling because it is easily accessible, whereas the site of primary infection in certain diseases may not be easily accessible [6]. Due to the ex vivo instability of RNA transcripts [1], a key challenge in gene expression analysis of blood samples is the rapid sample handling and stabilization of the mRNA by adding an RNA preservative (PAXgene^TM^ Blood RNA Tubes, Tempus^TM^ Blood RNA tubes, RNAlater Stabilization Reagent, RNAgard^®^ Blood Tubes). These four systems use proprietary reagents that are intended to stabilize RNA and ensure gene expression profiles that reflect the blood’s state at the moment of sampling. RNA stabilization technologies in blood collection tubes were developed almost 20 years ago, based on cell lysis and the inactivation of nucleases [7]. The latter stabilizes the RNA and prevents the induction of new transcripts [7,8]. Rapid addition or the direct withdrawal of blood into RNA preservatives is a major challenge in fieldwork in biomonitoring studies and is not even always possible. Recent studies have investigated gene instability in whole blood collected immediately in a blood tube versus at different bench times (hours [hr] or days), the collection with or without RNA preservatives or anticoagulants, and storage at different temperatures or for extended periods of time. In this context, bench time is considered the time between blood collection and sample stabilization, either through the addition of RNA preservatives or storage at low temperatures. Differences in gene expression were mainly observed between different bench times (2 h minimum), followed by anticoagulants, and to a much lesser extent storage temperatures and duration [9,10].

BioMole (Turin, Italy) has developed and, in 2023, officially launched on the market a novel blood stabilizer, called RNApro, in which RNA is stabilized during phlebotomy and sample storage. In this study, RNApro performance intended as RNA yield, integrity, and stability as evaluated. The second aim of this study focuses on the effect of RNApro conserved temperature on RNA yield, RNA quality, and specific gene expression.

## 2. Materials and Methods

### 2.1. Study Design and Subjects

After our institutional review board approved the study and obtained written informed consent from the subjects, forty samples of peripheral whole blood were collected according to the International Conference on Harmonization guidelines for good clinical practice as follows: Approximately 5 mL of blood was collected in two 4 mL Vacutainer Plus K2 EDTA tubes (Becton Dickinson, Franklin Lakes, NJ, USA), routinely used in the immunopathology laboratory of the Regina Margherita Hospital in Turin (Piedmont, Italy); (1) two aliquots of 200 μL peripheral blood were added to 800 μL of RNApro solution (stored at −80 °C until use) in a 1.5-mL Eppendorf tube and resuspended by vortexing; one aliquot was stored at −80 °C until use from 2017 to 2022 and one aliquot was used immediately; (2) 200 μL of peripheral blood was resuspended to 800 μL RNApro solution (stored at −80 °C until use) in 1.5 mL Eppendorf tubes, resuspended by shaking, and stored at −80 °C, −20 °C, 4 °C, and room temperature until use; (3) 200 μL of peripheral blood was resuspended to 800 μL RNApro solution (stored at −80 °C, −20 °C, 4 °C, and room temperature until use) in a 1.5 mL Eppendorf tube, resuspended by vortexing, and stored at −80 °C until use; (4) the remaining blood was collected directly into two 2.5 mL PAXgeneTM blood RNA tubes (PreAnalytiX, Qiagen BD, Valencia, CA, USA) and stored at −80 °C for 24 h before processing.

### 2.2. RNA Quality and Quantity

Total RNA was extracted using the RNA blood kit (automated Maxwell extractor from Promega, Madison, WI, USA), adapting the RNA blood kit protocol in cases 1–3 to use RNApro and using the manufacturer’s instructions for PAXgene tubes. In cases 1–3, we omitted the lysis of erythrocytes off-board and the use of the kit’s lysis buffer. We started directly with the addition of 1 mL of blood in RNApro in the cartridge of the kit. This kit provides for treatment with DNase during the first step of RNA extraction. To exclude contamination with residual genomic DNA, the total RNA extracts were amplified directly without reverse transcription. RNA concentration and purity were determined spectroscopically (ND-1000 spectrophotometer, Biochrom EnterpriseWaterbeach, Cambridge, UK) at an absorbance of 260 and 280 nm. An A260/A280 ratio of 1.8/2.1 is indicative of highly purified RNA. The RNAs were stored at −80° until use.

### 2.3. cDNA Synthesis

A total of 400 ng of RNA (20 μL of 20 ng/μL) was reverse-transcribed with 2 μL of buffer 10×, 4.8 μL of MgCl2 25 mM, 1 μL of RNase inhibitor 20U/l, 2 μL of ImpromII (Promega), 2 μL of mixed dNTPs 100 mM (Promega), 0.4 μL of random hexamers 250 μM (Promega), and dd-water in a final volume of 40 μL. The reaction mix was carried out in a GeneAmp PCR system, 9700 Thermal Cycle (Applied Biosystems, Foster City, CA, USA), under the following conditions: 5 min at 25 °C, 60 min at 42 °C, and 15 min at 70 °C for the inactivation of enzymes; the cDNAs were stored at −20 °C until use.

### 2.4. cDNA RT-qPCR

To assess the effect of the blood collection methods on gene expression, the isolated RNA samples were subjected to RT-qPCR analysis. Transcription levels of *GAPDH* were achieved as previously described in detail [11,12], using the primers and probes reported in Table 1. Briefly, 1/10 of cDNA was amplified in a 20 μL total volume reaction, containing 2.5 U of goTaQ MaterMix (Promega), 1.25 mmol/L of MgCl2, 200 nmol of specific probes, and 500 nmol of specific primers.

All amplifications were performed in a 96-well plate at 95 °C for 10 min, followed by 45 cycles at 95 °C for 15 s and at 60 °C for 1 min. Each sample was run in triplicate. All these tests were performed on the cfx96 real-time system (Bio-Rad Laboratories Segrate (MI), Italy) in combination with the Bio-Rad cfx Maestro software ver. 1.0.

### 2.5. DNA Contamination

In order to verify the absence of contaminating DNA in the extracted RNA, the latter was amplified directly without reverse transcription.

### 2.6. miRNA Amplification

miRNAs reverse transcription (starting from 500 ng of total RNA) was carried out with a Gene Amp RNA PCR kit (Life technologies, Carlsbad, CA, USA) including some modifications: 50 U of MMLV RT, 1 mM of dNTPs, 5 mM of MgCl2, 1 U of RNase Inhibitor, 1× PCR Buffer II, and 0,5 µg of specific SLP (Table 1). The reaction was performed with an initial incubation at 16 °C for 30 min followed by a second step at 42 °C for 1 h. In order to end the reverse transcription step, a final incubation at 99 °C for 5 min was performed. After the reverse transcription step, an asymmetric PCR using 300 nM of specific forward primer (Table 2), 0,1 U of GoTaq^®^ Hot Start polymerase (Promega, Bergamo, Italy), 4 μL of 5X Colorless GoTaq^®^ Flexi Buffer, and 2 μL of cDNA, obtaining a final volume of 20 μL, was carried out. The thermal profile used was 95 °C for 2 min; 30 cycles of 94 °C for 15 s, 55 °C for 30 s, and 72 °C for 20 s. A total of 5 μL of enriched cDNA, denominated ccDNA, was added to 35 μL of reaction mix containing 800 nM of forward primer, 1000 nM of Universal Reverse primer, 200 nM of MGB probe, and 1× TaqMan Universal PCR Master Mix (P/N: 4324018 Life Technologies, Carlsbad, CA, USA) in a final volume of 40 μL.

### 2.7. Statistical Analysis

The Wilcoxon test was used to compare RNA quantity and quality and the *GAPDH* ct of the amplification of samples extracted from 2017 to 2023 and to compare RNA quantity and *GAPDH* ct between samples extracted after RNApro supplementation and a PAXgene tube.

The Freidman test and the Wilcoxon test were used to compare RNA quantity and the *GAPDH* ct of the amplification of RNApro and blood supplemented with RNApro, stored at −80 °C, −20 °C, 4 °C, and room temperature.

Statistical analyses were performed using the Prism software version 7 (GraphPad Software, La Jolla, CA, USA). In all analyses, *p* < 0.05 was taken to be statistically significant.

## 3. Results

### 3.1. RNA Yield

The tubes containing the peripheral blood and RNApro stored from 2017 to 2023 were extracted. All samples were extracted twice, once in accordance with the dates of storage and once in 2023, and the results of yield, purity, and gene expression were reported in Figure 1, Figure 2 and Figure 3.

The RNA quantity in blood stored in 2017 and extracted after 48 h at −80 °C and extracted in 2023 after 6 years of incubation at −80 °C were (median age and 25–75% IQR) were 28 ng/μL, 25–32.25 and 28 ng/μL, 22.75–36.25, respectively. The RNA quantity was not significantly different between the blood stored in 2017 and that extracted after 48 h of incubation at −80 °C and that extracted in 2023 after 6 years of incubation at −80 °C (*p* = 0.8340), as shown in Figure 1.

The RNA quantity in blood stored in 2018 and extracted after 48 h at −80 °C and extracted in 2023 after 5 years of incubation at −80 °C were (median age and 25–75% IQR) 29 ng/μL, 22–37.5 and 33.5 ng/μL, 21–37.27, respectively. The RNA quantity was not significantly different between blood stored in 2018 and extracted after 48 h of incubation at −80 °C and extracted in 2023 after 5 years of incubation at −80 °C (*p* = 0.4697), as shown in Figure 1. 

The RNA quantity in blood stored in 2019 and extracted after 48 h at −80 °C and extracted in 2023 after 4 years of incubation at −80 °C were (median age and 25–75% IQR) 23.05 ng/μL, 21.77–30.77 and 25.55 ng/μL, 21.85–33.5, respectively. The RNA quantity was not significantly different between blood stored in 2019 and extracted after 48 h of incubation at −80 °C and extracted in 2023 after 4 years of incubation at −80 °C (*p* = 0.0677), as shown in Figure 1.

The RNA quantity in blood stored in 2020 and extracted after 48 h at −80 °C and extracted in 2023 after 3 years of incubation at −80 °C were (median age and 25–75% IQR) 25.8 ng/μL, 18.8–31.25 and 26.65 ng/μL, 21.37–33.77, respectively. The RNA quantity was not significantly different between blood stored in 2020 and extracted after 48 h of incubation at −80 °C and extracted in 2023 after 3 years of incubation at −80 °C (*p* = 0.4824), as shown in Figure 1.

The RNA quantity in blood stored in 2021 and extracted after 48 h at −80 °C and extracted in 2023 after 2 years of incubation at −80 °C were (median age and 25–75% IQR) 25.4 ng/μL, 20.4–34.37 and 25.5 ng/μL, 16.77–31.62, respectively. The RNA quantity was not significantly different between blood stored in 2021 and extracted after 48 h of incubation at −80 °C and extracted in 2023 after 2 years of incubation at −80 °C (*p* = 0.0951), as shown in Figure 1.

The RNA quantity in blood stored in 2022 and extracted after 48 h at −80 °C and extracted in 2023 after 1 year of incubation at −80 °C were (median age and 25–75% IQR) 27.25 ng/μL, 24.6–32.27 and 25.25 ng/μL, 22.67–33.5, respectively. The RNA quantity was not significantly different between blood stored in 2022 and extracted after 48 h of incubation at −80 °C and extracted in 2023 after 1 year of incubation at −80 °C (*p* = 0.1124), as shown in Figure 1.

**Figure 1 diagnostics-14-00971-f001:**
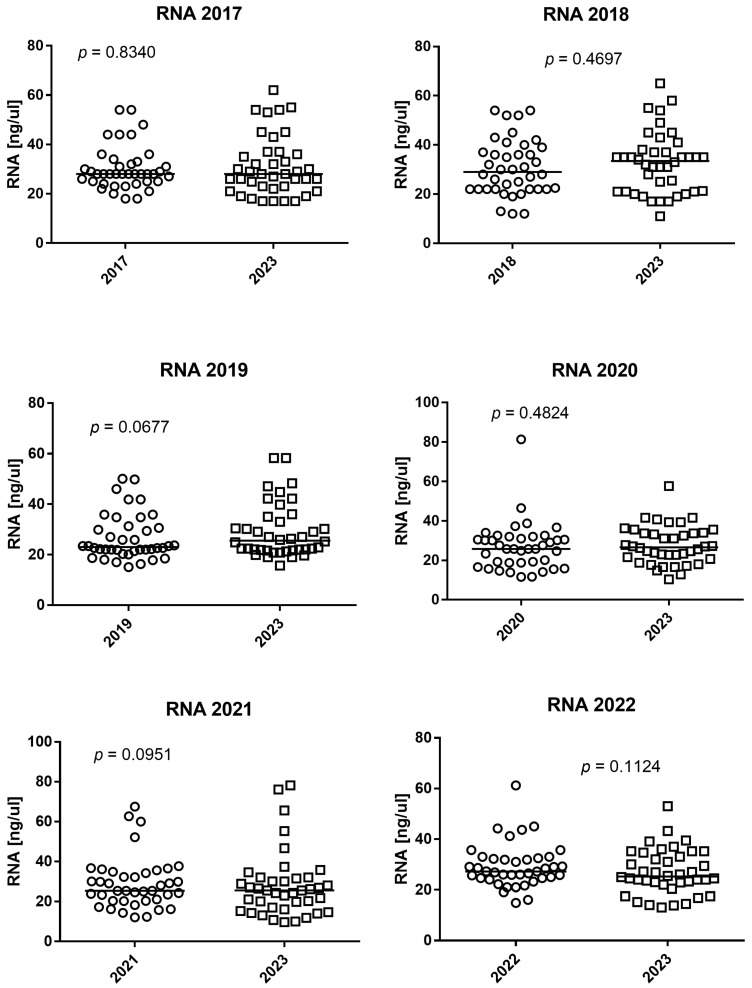
Wilcoxon test of spectrophotometric RNA yield expressed in ng/μL. The horizontal line in the chart represents the median of the values.

### 3.2. RNA Quality

The RNA quality in blood stored in 2017 and extracted after 48 h at −80 °C and extracted in 2023 after 6 years of incubation at −80 °C were (median age and 25–75% IQR) 2 R260/280, 1.9–2.01 and 2 R260/280, 1.96–2, respectively. The RNA quality was not significantly different between blood stored in 2017 and extracted after 48 h of incubation at −80 °C and extracted in 2023 after 6 years of incubation at −80 °C (*p* = 0.8123), as shown in Figure 2. 

**Figure 2 diagnostics-14-00971-f002:**
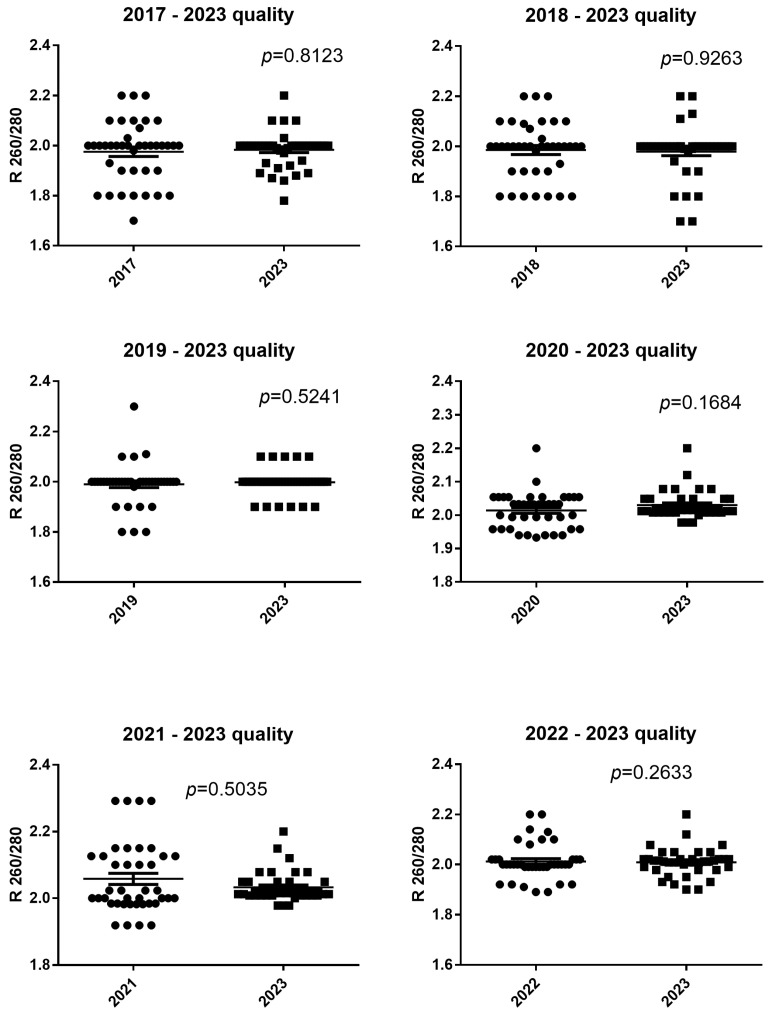
Wilcoxon test of spectrophotometric RNA purity expressed as R 260/280 nm. The horizontal line in the chart represents the median of the values.

The RNA quality in blood stored in 2018 and extracted after 48 h at −80°C and extracted in 2023 after 5 years of incubation at −80°C were (median age and 25–75% IQR) 2 R260/280, 1.9–2.04 and 2 R260/280, 1.99–2, respectively. The RNA quality was not significantly different between blood stored in 2018 and extracted after 48 h of incubation at −80 °C and extracted in 2023 after 5 years of incubation at −80 °C (*p* = 0.9263), as shown in Figure 2. 

The RNA quality in blood stored in 2019 and extracted after 48 h at −80 °C and extracted in 2023 after 4 years of incubation at −80 °C were (median age and 25–75% IQR) 2 R260/280, 2-2 and 2 R260/280, 2-2, respectively. The RNA quality was not significantly different between blood stored in 2019 and extracted after 48 h of incubation at −80 °C and extracted in 2023 after 4 years of incubation at −80 °C (*p* = 0.5241), as shown in Figure 2.

The RNA quality in blood stored in 2020 and extracted after 48 h at −80 °C and extracted in 2023 after 3 years of incubation at −80 °C were (median age and 25–75% IQR) 2.03 R260/280, 1.96–2.05 and 2.01 R260/280, 2.01–2.05, respectively. The RNA quality was not significantly different between blood stored in 2020 and extracted after 48 h of incubation at −80 °C and extracted in 2023 after 3 years of incubation at −80 °C (*p* = 0.1684), as shown in Figure 2.

The RNA quality in blood stored in 2021 and extracted after 48 h at −80 °C and extracted in 2023 after 2 years of incubation at −80 °C were (median age and 25–75% IQR) 2.01 R260/280, 1.98–2.13 and 2.01 R260/280, 2.01–2.05, respectively. The RNA quality was not significantly different between blood stored in 2021 and extracted after 48 h of incubation at −80 °C and extracted in 2023 after 2 years of incubation at −80 °C (*p* = 0.5035), as shown in Figure 2.

The RNA quality in blood stored in 2022 and extracted after 48 h at −80 °C and extracted in 2023 after 1 year of incubation at −80 °C were (median age and 25–75% IQR) 2 R260/280, 1.99–2.02 and 2.01 R260/280, 1.99–2.02, respectively. The RNA quality was not significantly different between blood stored in 2022 and extracted after 48 h of incubation at −80 °C and extracted in 2023 after 1 year of incubation at −80 °C (*p* = 0.2633), as shown in Figure 2.

The RNA amplificability in blood stored in 2017 and extracted after 48 h at −80 °C and extracted in 2023 after 6 years of incubation at −80 °C were (median age and 25–75% IQR) 22.13 Ct, 21.57–22.75 and 22.61 Ct, 21.48–23.52, respectively. The RNA quantity was not significantly different between blood stored in 2017 and extracted after 48 h of incubation at −80 °C and extracted in 2023 after 6 years of incubation at −80 °C (*p* = 0.1648), as shown in Figure 3.

**Figure 3 diagnostics-14-00971-f003:**
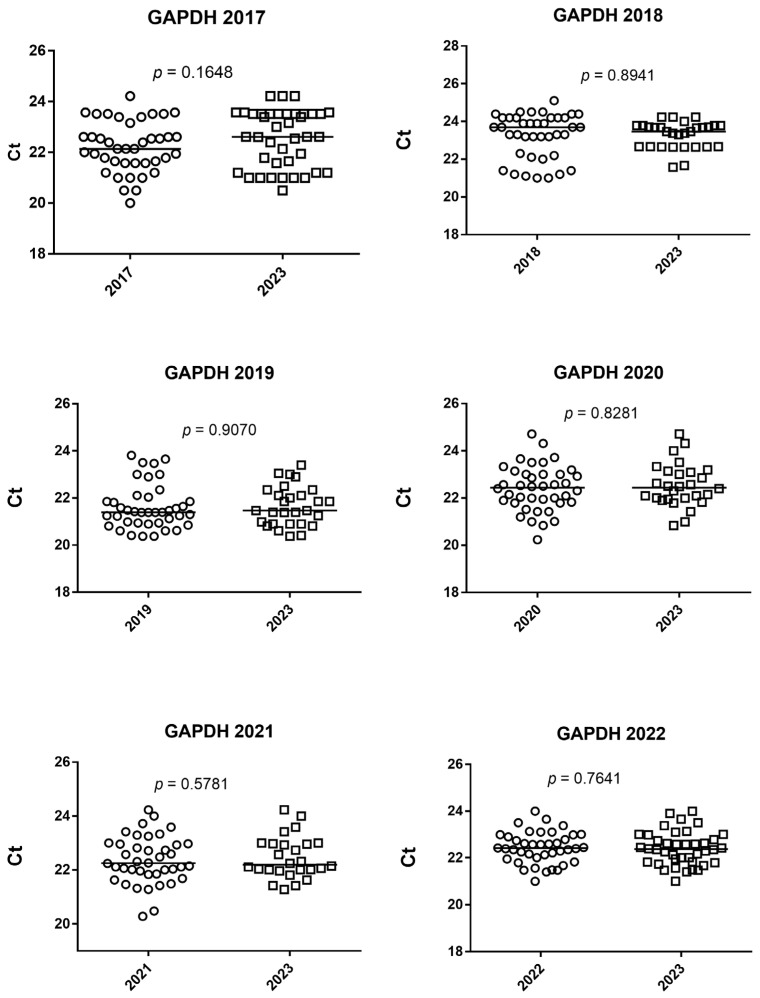
Wilcoxon test of GAPDH gene expression expressed as Ct of amplification. The horizontal line in the chart represents the median of the values.

The RNA amplificability in blood stored in 2018 and extracted after 48 h at −80 °C and extracted in 2023 after 5 years of incubation at −80 °C were (median age and 25–75% IQR) 23.70 Ct, 22.27–24.2 and 23.46 Ct, 22.66–23.70, respectively. The RNA quantity was not significantly different between blood stored in 2018 and extracted after 48 h of incubation at −80 °C and extracted in 2023 after 5 years of incubation at −80 °C (*p* = 0.8941), as shown in Figure 3.

The RNA amplificability in blood stored in 2019 and extracted after 48 h at −80 °C and extracted in 2023 after 4 years of incubation at −80 °C were (median age and 25–75% IQR) 21.39 Ct, 20.94–21.89 and 21.47 Ct, 20.90–22.06, respectively. The RNA quantity was not significantly different between blood stored in 2019 and extracted after 48 h of incubation at −80 °C and extracted in 2023 after 4 years of incubation at −80 °C (*p* = 0.9070), as shown in Figure 3.

The RNA amplificability in blood stored in 2020 and extracted after 48 h at −80 °C and extracted in 2023 after 3 years of incubation at −80 °C were (median age and 25–75% IQR) 22.44 Ct, 21.81–23 and 22.35 Ct, 21.88–23.02, respectively. The RNA quantity was significantly different between blood stored in 2020 and extracted after 48 h of incubation at −80 °C and extracted in 2023 after 3 years of incubation at −80 °C (*p* = 0.8281), as shown in Figure 3.

The RNA amplificability in blood stored in 2021 and extracted after 48 h at −80 °C and extracted in 2023 after 2 years of incubation at −80 °C were (median age and 25–75% IQR) 22.25 Ct, 21.84–22.96 and 22.19 Ct, 21.82–22.89, respectively. The RNA quantity was not significantly different between blood stored in 2021 and extracted after 48 h of incubation at −80 °C and extracted in 2023 after 2 years of incubation at −80 °C (*p* = 0.5781), as shown in Figure 3.

The RNA amplificability in blood stored in 2022 and extracted after 48 h at −80 °C and extracted in 2023 after 1 year of incubation at −80 °C were (median age and 25–75% IQR) 22.1 Ct, 21.–24 and 22.3 Ct, 21–23.7, respectively. The RNA quantity was not significantly different between blood stored in 2022 and extracted after 48 h of incubation at −80 °C and extracted in 2023 after 1 year of incubation at −80 °C (*p* = 0.7641), as shown in Figure 3.

### 3.3. Sample Storage Temperature

To determine the optimal storage temperature for peripheral blood samples with the addition of RNApro, the tubes were incubated at −80 °C, −20 °C, 4 °C, and room temperature for one week or one month. The RNA was then extracted and the concentration and amplificability were evaluated (Figure 4 and Figure 5).

The amount of RNA in the tubes incubated at −80 °C, −20 °C, 4 °C, and room temperature for one week was (mean age and 25–75% IQR) 25.4 ng/μL, 21.7–30.25; 24.6 ng/μL, 22–29.9; 25 ng/μL, 19.95–28.2; and 23.9 ng/μL, 20–28.5. The amount of RNA dropped significantly for samples stored with RNApro at room temperature (*p* = 0.0150), as shown in Figure 4.

The RNA quantity in tubes incubated at −80 °C, −20 °C, 4 °C, and room temperature for one month was (median age and 25–75% IQR): 25 ng/μL, 21.55–29.25; 24.7 ng/μL, 21–25.95; 23 ng/μL, 21–28.2; and 15 ng/μL, 12–21.9, respectively. The amount of RNA dropped significantly for samples stored with RNApro at room temperatures (*p* = 0.0203), as shown in Figure 4. In particular, the storage temperature of the samples does not seem to influence up to 4 °C. At room temperature, the extraction yield noticeably worsens.

**Figure 4 diagnostics-14-00971-f004:**
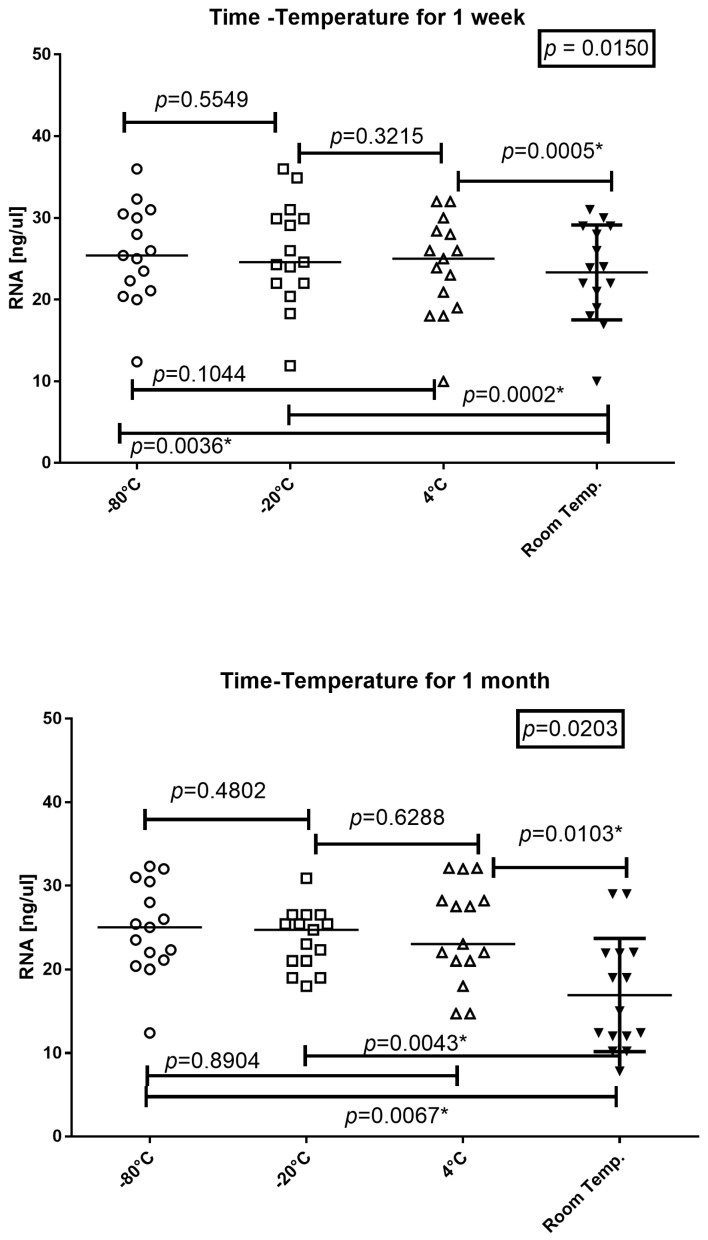
Friedman test and Wilcoxon test of spectrophotometric RNA yield expressed in ng/μL for tubes incubated at −80 °C, −20 °C, 4 °C, and room temperature for one week and/or one month. The horizontal line in the chart represents the median of the values; * = *p* value < 0.05.

RNA-GAPDH expression levels in tubes incubated for one week at −80 °C, −20 °C, 4 °C, and room temperature were (mean age and 25–75% IQR) 21.51 ng/μL, 21.37–21.97; 21.46 ng/μL, 21.28–21.90; 21.59 ng/μL, 21.55–22.11; and 23.80 ng/μL, 23.76–24.30. The expression levels of *GAPDH* differ significantly between blood stored with RNApro at different temperatures (*p* < 0.0001), as shown in Figure 5. In particular, storage at room temperature always leads to an increase in Ct data. The Wilcoxon test is always statistically significant when comparing room temperature with −80 °C, −20 °C, and 4 °C, with *p* < 0.0001.

RNA-GAPDH expression levels in tubes incubated at −80 °C, −20 °C, 4 °C, and room temperature for one month were (mean age and 25–75% IQR) 22.28 ng/μL, 21.75–22.31; 22.1 ng/μL, 21.51–22.49; 23.64 ng/μL, 23.07–24.10; and 32.97 ng/μL, 32.26–34.10. The expression levels of *GAPDH* differ significantly between blood stored with RNApro at different temperatures (*p* < 0.0001), as shown in Figure 5. In particular, only storage at −80 °C and −20 °C shows no difference, *p* = 0.5817. All other combinations are statistically significant with *p* < 0.0001.

**Figure 5 diagnostics-14-00971-f005:**
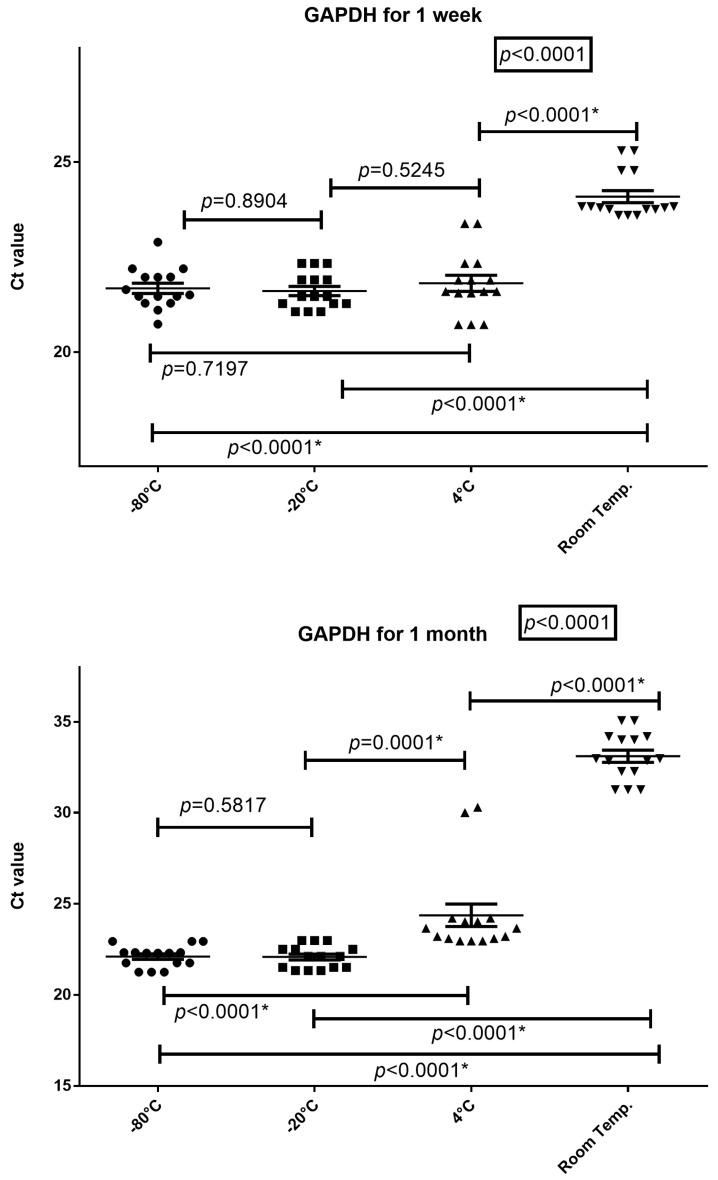
Friedman test and Wilcoxon test of GAPDH RNA expression levels for tubes incubated at −80 °C, −20 °C, 4 °C, and room temperature for one week and/or one month. The horizontal line in the chart represents the median of the values; * = *p* value < 0.05.

### 3.4. RNApro Store Temperature

To determine the optimal storage temperature for RNApro, peripheral blood samples were spiked with RNApro that had previously been incubated at −80 °C, −20 °C, 4 °C, and room temperature for two months. The RNA was then extracted and the concentration and amplificability were assessed (Figure 6).

RNApro was stored at −80 °C, −20 °C, 4 °C, and room temperature before peripheral blood was added. The amount of RNA was (mean age and 25–75% IQR) 24.3 ng/μL, 20.05–29.18; 23 ng/μL, 15.9–27.25; 22.4 ng/μL, 15.23–28.5; and 20.2 ng/μL, 15.83–26.6, respectively. 

The amount of RNA dropped for blood extracted with RNApro stored at room temperature (*p* = 0.0045), as shown in Figure 6.

The amplificability of RNA was (mean age and 25–75% IQR): 21.92 Ct, 21.05–22.16; 21.98 Ct, 21.47–22.21; 21.91 Ct, 21.32–22.19; and 21.91 Ct, 21.31–22.19, respectively. The amplificability of RNA was not significantly different between blood extracted with RNApro and stored at different temperatures (*p* = 0.4348), as shown in Figure 6.

**Figure 6 diagnostics-14-00971-f006:**
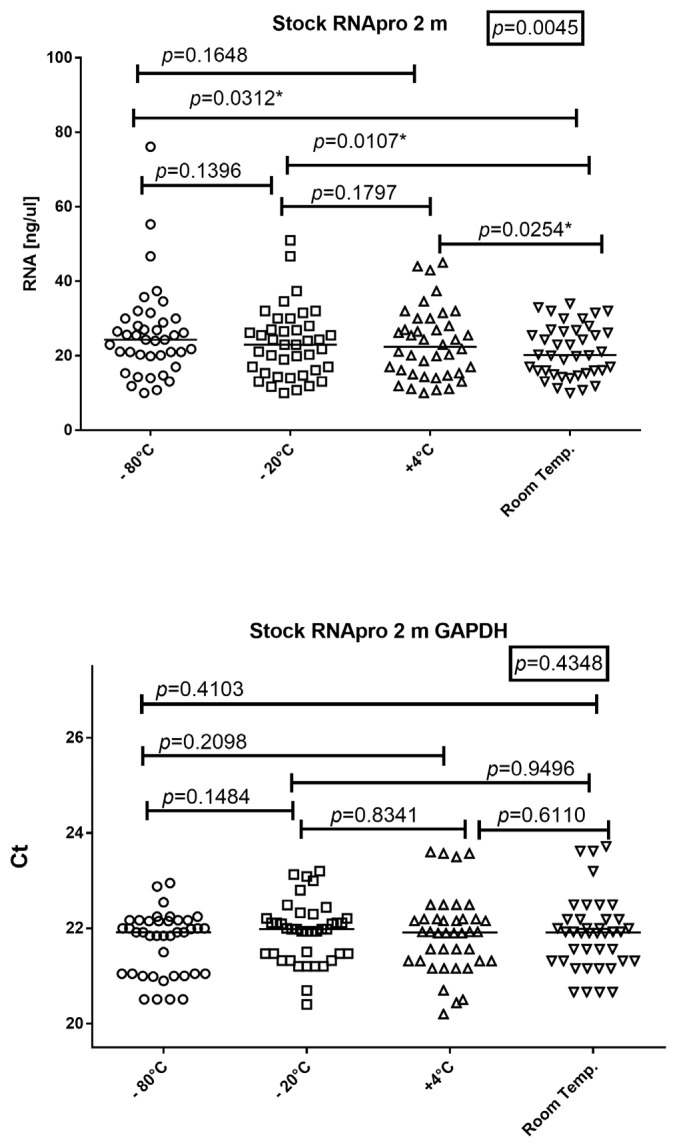
Friedman test and Wilcoxon test results for RNA quantity and GAPDH RNA expression levels obtained from RNApro previously stored at −80 °C, −20 °C, 4 °C, and room temperature. The horizontal line in the chart represents the median of the values; * = *p* value < 0.05.

### 3.5. Comparison between RNApro and PAXgene

For comparison between the RNApro stabilizer and the actual validated commercial PAXgene tube, ten peripheral blood samples were preserved in parallel and extracted with a commercial automated RNA kit as recommended by the manufacturer. RNApro and PAXgene were stored at −80 °C and room temperature until use. The amount of RNA was (mean age and 25–75% IQR) 25.7 ng/μL, 23.23–30.3 for RNApro and 18.15 ng/μL, 15.3–18.55 for PAXgene, respectively. The amount of RNA was significantly different between blood extracted with RNApro and PAXgene (*p* = 0.0020), as shown in Figure 7. The quality of RNA was (mean age and 25–75% IQR) 1.96 R260/280, 1.91–2 for RNApro and 1.70 R260/280, 1.63–1.77 for PAXgene. The quality of RNA was significantly different between blood extracted with RNApro and PAXgene (*p* = 0.0156), as shown in Figure 7. The amplificability of RNA was (mean age and 25–75% IQR) 22.68 Ct, 22.11–23.06 for RNApro and 29.60 Ct, 28.03–29.87 for PAXgene. The quality of RNA was significantly different between blood extracted with RNApro and PAXgene (*p* = 0.0020), as shown in Figure 7.

### 3.6. DNA Contamination

The RNA relative to the experiment reported in Figure 6 was also amplified directly without retrotranscription. No signal or Ct > 40 was observed in any of the samples.

### 3.7. miRNA Amplification

To determine the amplificability of miRNA, RNA43 was amplified in 40 samples of peripheral blood stored at −80 °C with RNApro. Ct was obtained for all samples tested. The amplificability of RNA was (mean age and 25–75% IQR) 12.57 Ct, 12.53–14.28. 

## 4. Discussion

High-quality genetic and genomic material is an important resource for human studies. Human blood has been recognized as an important diagnostic resource for centuries. Blood is a complex fluid that is in constant contact with all body tissues, thereby providing information from a variety of unique compartments that include nucleated white blood cells (WBCs), nucleated red blood cells (RBCs), and cell-free RNA: ribonucleoprotein complexes and additional vesicular debris from various tissues in the body [13]. As blood sampling is considered a non-invasive procedure, it is often used for the assessment of a range of disease-related biomarkers. In addition, the increasing application of personalized medicine in the treatment of chronic diseases has shown that RNA signatures can be used to specifically optimize the most appropriate treatment strategies for the patient. This prompted the development of a variety of unique methods for the collection, stabilization, and extraction of RNA from blood. After collection, nucleic acids in the blood are susceptible to oxidative damage and nuclease attack. In addition, the relatively unstable intracellular RNAs are subject to both transcript induction and transcript degradation, resulting in altered gene expression ex vivo. Taken together, these experimental variations make it difficult to detect true biological variance. Different phases after blood collection, including blood collection, transportation, and storage, as well as processing steps such as the RNA isolation method, can influence gene expression profiles. Therefore, the optimization of these technical variables is necessary to reveal the true biological patterns in the blood. Although new commercial systems for the collection and rapid stabilization of RNA from peripheral blood continue to enter the market, there is insufficient systematic updating of research showing the impact of these solutions on whole-genome profiling. The collection and preservation of RNA, which is particularly susceptible to degradation, has historically been limited by the ability to rapidly isolate or freeze the sample. The most important chemical component for the extraction of RNA is guanidinium thiocyanate (GTC) [14]. The main component of RNApro is GTC. GTC is a strong denaturant that can inhibit RNase and DNase activities and separate nucleic acids and nucleoproteins, thereby preserving the integrity of RNA and DNA [14]. Several studies have shown that neither DNA nor RNA are adsorbed to the surface of magnetic beads if the lysis buffer does not contain GTC [14]. The automation of conventional nucleic acid extraction methods is difficult to implement due to the centrifugal steps [15]. The development of magnetic bionanoparticles has greatly promoted the automation of nucleic acid extraction [14]. Numerous studies have shown that magnetic beads are capable of absorbing DNA and RNA [16,17], and their adsorption capacity can be significantly enhanced through appropriate concentrations of some solvents, including GTC. The reasons for this enhancement may be as follows [18,19]: anions are present on the surfaces of the magnetic beads and the nucleic acids, and GTC or ammonium can act as a bridge between the nucleic acids and the magnetic beads [20]; the high concentration of salt ions dehydrates the water layer on the surfaces of both the nucleic acids and the magnetic beads by forming hydrated ions, which cause the magnetic beads to adsorb nucleic acids; and GTC makes the double-stranded DNA single-stranded because the basic group of the single-stranded DNA and the hydroxyl group on the surface of the magnetic beads form a chemical bond that enhances adhesion. Therefore, DNA and RNA are barely adsorbed onto the magnetic bead surface without GTC. RNApro is an RNA stabilization reagent based on GTC such as PAXgene^TM^ Blood RNA Tubes, Tempus^TM^ Blood RNA tubes, RNAgard RNA tubes, and RNAlater Stabilization Reagent. Our results show that blood samples stored at −80 °C and re-extracted after 7 years show no differences in terms of quantity, quality, and amplifiability. This result was also previously shown for PAXgene tubes in the PAXgene^®^ Blood RNA System Technical Note [21], demonstrating the stability of RNA in blood samples stored at −20 °C and −70 °C for 11 years (132 months) and for Tempus tubes for a period of 6 years [22]. PAXgene, Tempus tubes, and RNAgard blood tubes were marketed in a fixed format and a large volume of blood sample (3–5 mL) must be added to the tube before storing or extracting the RNA and not offer the Stabilizing Reagent that comes inside the tube as a stand-alone item. The total RNA yield was affected by different preanalytical variables in the two systems: the filling level of the blood collection tube significantly affected the yield in the PAXgene, but not in the Tempus system, while suboptimal tube inversions significantly affected the yield in the Tempus, but not in the PAXgene system. Differences in the behavior of the two types of tubes may be explained by different stabilization mechanisms: chaotropic agents or micelles formation [23]. RNApro is marketed as an RNA stabilizer that can be added to samples in a 5:1 ratio in a second step. Stellino and colleagues demonstrated the robustness of some common preanalytical variables, such as the lack of tube inversion and the suboptimal tube filling impact on subsequent RNA extraction and RNA yield, quality, and integrity [23]. Using RNApro, we can choose the volume of blood to store each time based on needs. This fact is very important when we are dealing with pediatric or neonatal patients.

Tempus Blood RNA Tubes, PAXgene, RNAgard, RNAlater Stabilization Solution, and RNApro all use the same effective stabilization of mRNA expression profiles and eliminate the need to isolate the RNA immediately after sample collection. However, there are some key differences: (1) The components in each stabilization solution are different. The exact components and differences are proprietary. (2) The RNAlater stabilization solution preserves the RNA from degradation without lysing the blood cells. The samples in the RNAlater stabilization solution can be stored at room temperature for up to one week or at 4 °C for up to one month. Similar results can also be observed for PAXgene tubes, Tempus tubes, and RNAgard tubes [24]. In agreement with these data, the RNApro stabilization solution preserves the RNA from degradation for up to 1 month at 4 °C and 1 week at room temperature. RNApro can be stored indifferently at −80, −20, 4 °C, or room temperature for up to 2 months, and then could be stored at −80 °C for up to seven years. These data are in accordance with other RNA stabilizer reagents.

Previous data show that gene expression differences between PAXgene and Tempus blood RNA tubes are very reproducible between independent samples and biobanks [25]. We decided to compare the RNApro stabilizer with the PAXgene tube. Although the data available to us are limited and were obtained using an RNA extraction method “adapted” to PAXgene, which differs from the original protocol intended by the manufacturer, the amplification and recovery data were statistically better when the blood samples had been, stored with RNApro. Of course, we cannot say that RNApro is better than PAXgene tubes, which would require another study dedicated solely to this purpose, but the performance of the product is certainly very encouraging when used on a large scale. At the transcriptional level, there is no perfect overlap of data using different stabilization platforms [26]. Donohue et al. compared the gene expression profiles of engineered replicate samples collected with PAXgene and RNAgard tubes and found large differences between samples from the same individual when these two different systems were used. The results of some studies suggest the use of a single preservation platform throughout the course of a given study, without mixing or switching collection methods during sample collection and preservation [26]. These observations strongly caution against the direct comparison of results from samples preserved in RNAgard^®^ with those preserved in PAXgene, Tempus tubes, or RNAlater as they may give rise to methodologically related false-positive results.

We have also shown that the RNApro stabilizer is also suitable for the preservation and subsequent extraction of miRNAs, in agreement with the other stabilizers [23,27,28,29], but we did not compare the data with other RNA stabilization methods. Better characterization of the effects of collection method idiosyncrasies will facilitate further research into understanding the effect of gene expression on variability in human sample types. These results point out the need for a strict standardization of handling the blood specimen with regard to peripheral blood sample processing time between phlebotomy and RNA isolation. 

In summary, our study is the first to analyze the performance of an RNA stabilizer called RNApro. RNA was extracted using an automated extractor and its quantity, purity, and amplifiability were evaluated. As an example, the RNA stabilized with RNApro was compared with the RNA stabilized in PAXgene. We can conclude that several studies have shown significant differences in gene expression analysis when the sample was preserved in different RNA stabilizers. Therefore, it is desirable to standardize the method of nucleic acid conservation when comparing data from transcriptomic analyses.

## Figures and Tables

**Figure 7 diagnostics-14-00971-f007:**
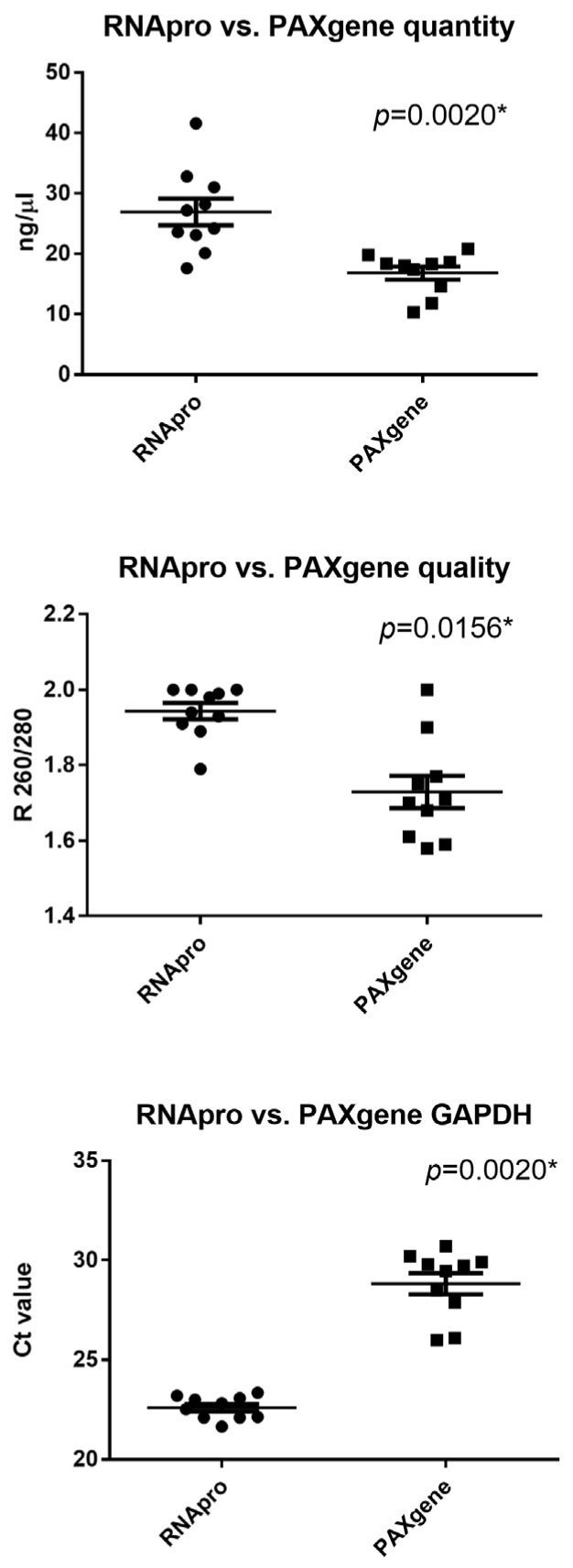
Wilcoxon test of spectrophotometric RNA yield, expressed in ng/μL, and of spectrophotometric RNA purity, expressed as R 260/280 nm, obtained with RNApro stabilizer and PAXgene tubes. The horizontal line in the chart represents the median of the values; * = *p* value < 0.05.

**Table 1 diagnostics-14-00971-t001:** Primers and probes used to assess the transcription levels of *GAPDH*. Amplicon length of 169 bp.

Name	Primer/ Probe	Sequence
*GAPDH*	Forward	5′-CGAGATCCCTCCAAAATCAA-3′
	Reverse	5′-TTCACACCCATGACGAACAT-3′
	Probe	6FAM-5′-TCCAACGCAAAGCAATACATGAAC-3′-TAMRA

**Table 2 diagnostics-14-00971-t002:** Primers and probes list and miRNAs target sequences. All sequences are written in 5′-3′ direction.

Target	RNU43
Sequence	GAACUUAUUGACGGGCGGACAGAAACUGUGUGCUGAUUGUCACGUUCUGAUU
SLP	GGCTCTGGTGCAGGGTCCGAGGTATTCGCACCAGAGCCAATCAG
Forward	TGACGGGCGGACAGAAA
Probe MGB fam	TGTGTGCTGATTGTCA
Universal reverse primer	TGCAGGGTCCGAGGTATTC

## Data Availability

The data that support the findings of this study are available from the corresponding author upon reasonable request.

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
