# Peer review of "Characteristics of RNA Stabilizer RNApro for Peripheral Blood Collection"

_diagnostics, 2024, doi:10.3390/diagnostics14100971_

Round 1

Reviewer 1 Report

Comments and Suggestions for Authors

Dear Editor,

Gambarino et al. provided an interesting manuscript entitled “Characteristics of RNA Stabilizer RNApro for Peripheral Blood Collection” in which the authors described a new RNA stabilizer product.

The manuscript is interesting, with great potential clinical and practical value. However, I have some comments/suggestions that should be addressed before acceptance:

·        The authors used Mann-Whitney test to compare RNA concentrations at initial sampling time versus after several years. However, in this case, the data is paired and should be analysed with Wilcoxon test.

·        The authors used one-way ANOVA test, where it should be repeated measure ANOVA. The authors used again Mann-Whitney to compare groups after ANOVA was performed. This is also incorrect, and the appropriate post-hoc test should be performed instead of Mann-Whitney.

·        The authors performed Mann-Whitney test which is used for non-parametric data and then used ANOVA which is used for parametric data. Is the data parametric or not??

·        The materials and methods section should be between introduction and results since this is the standard MDPI structure.

·        There are unlabelled figures.

Author Response

  • The authors used Mann-Whitney test to compare RNA concentrations at initial sampling time versus after several years. However, in this case, the data is paired and should be analysed with Wilcoxon test.

We reanalyzed data of Figure 1, 2 and 3 with Wilcoxon test, as suggested.

  • The authors used one-way ANOVA test, where it should be repeated measure ANOVA. The authors used again Mann-Whitney to compare groups after ANOVA was performed. This is also incorrect, and the appropriate post-hoc test should be performed instead of Mann-Whitney.

We reanalyzed data of Figure 4, 5 and 6 with Friedman test and Wilcoxon test, as suggested.

  • The authors performed Mann-Whitney test which is used for non-parametric data and then used ANOVA which is used for parametric data. Is the data parametric or not??

The data is not parametric. We reanalysed all data with the correct tests.

  • The materials and methods section should be between introduction and results since this is the standard MDPI structure.

      We inserted Materials and Methods in the right position.

  • There are unlabelled figures.

We added labels to all figures.

Reviewer 2 Report

Comments and Suggestions for Authors

The authors have tested the performance of RNApro from liquid biopsies (including 7 year old -80C samples). They have studied the yield, integrity plus gene expression.  The article describes sufficient background in the literature and reproducible methods. They have prepared informative figures and presented a clear argument. 

However, I'm not convinced that there is no genomic DNA in the extracted RNA. They have used a -RT control for cDNA preparation but have not showed any data to support no amplification products on an agarose gel, or a low Ct value or no peaks on a melt curve profile.  The RTqPCR primers that they have used target an exonic region of GADPH. Thus, removal of DNA is a must. 

The authors should also include RNA integrity n# to support their spectrophotometer data. This is an added cost but I always use this prior to RNA seq since the spectrophotometer is not reliable to show intact RNA and absence of DNA.  Alternatively, amplification of a larger product size (5-10 kb) using standard RT-PCR and agarose gel to show that the RNA is intact.

Thus, additional work is required to support publication of this data with the current level of expectations for gene expression studies. 

Minor comments:

Line 37 - remove "recently named" as this has been known since 2020.

Line 70 - use RTqPCR instead of qPCR.

Line 93 - included when RNApro was released on the market as you give the impression that is a new product. 

Figure 1 - does the horizontal line in the chart that is embedded in circles or square the median or mean. If so, please add this to the figure legend.  

Line 431

Sentence does not make sense - " erythrocytes from board" .

Line 438 - cDNA

I don't understand how you can use 400 ng of RNA as a starting template in 20 uL. You only have 7.8 uL volume for RNA and your average was approx 28 ng/uL. Please check the right volumes were used or did you concentration your RNA.

Line 448

Use RTqPCR.  

Use italics for gene names - GADPH

Line 450

How did you measure cDNA concentration?  Best to use volume of cDNA used.

Table 1

Include product size - can be in the figure legend. 

Author Response

However, I'm not convinced that there is no genomic DNA in the extracted RNA. They have used a -RT control for cDNA preparation but have not showed any data to support no amplification products on an agarose gel, or a low Ct value or no peaks on a melt curve profile.  The RTqPCR primers that they have used target an exonic region of GADPH. Thus, removal of DNA is a must.

The RNA was  amplified directly without retrotranscription. No signal or Ct >40 was observed in any of the samples. We add this sentence in the text.

The authors should also include RNA integrity n# to support their spectrophotometer data. This is an added cost but I always use this prior to RNA seq since the spectrophotometer is not reliable to show intact RNA and absence of DNA.  Alternatively, amplification of a larger product size (5-10 kb) using standard RT-PCR and agarose gel to show that the RNA is intact.

I understand your doubts. We have decided to focus on amplifiability as a guarantee of the "usability" of the extracted RNA in a qPCR experiments. Our goal is to evaluate the integrity of the RNA(usable also in RNAseq experiment) in further work by also adding the fragmentation index expressed as long amplicon/short amplicon. The journal gave us 10 days to provide the reports and it is not possible to add this data now. I hope that the PCR amplifiability is sufficient for you at the moment.

Minor comments:

Line 37 - remove "recently named" as this has been known since 2020.

We remove it.

Line 70 - use RTqPCR instead of qPCR.

We change it.

Line 93 - included when RNApro was released on the market as you give the impression that is a new product.

We add a sentence.

Figure 1 - does the horizontal line in the chart that is embedded in circles or square the median or mean. If so, please add this to the figure legend. 

We add in the figure legend the sentence as recommended.

Line 431 Sentence does not make sense - " erythrocytes from board" .

We correct it.

Line 438 - cDNA

I don't understand how you can use 400 ng of RNA as a starting template in 20 uL. You only have 7.8 uL volume for RNA and your average was approx 28 ng/uL. Please check the right volumes were used or did you concentration your RNA.

I don't understand the calculation you made. The real volume of RNA extracted is 50 ul. All samples are diluted to a concentration of 20 ng/ul. In 20 ul there are therefore approximately 400 ng.

Line 448  Use RTqPCR. 

We do it.

Use italics for gene names - GADPH

We do it.

Line 450 How did you measure cDNA concentration?  Best to use volume of cDNA used.

We do it.

Table 1: Include product size - can be in the figure legend.

We add it.

Round 2

Reviewer 1 Report

Comments and Suggestions for Authors

The manuscript was improved and should be accepted for publication.

Author Response

thanks for tje comments.

Reviewer 2 Report

Comments and Suggestions for Authors

The authors have addressed all my comments.  They have made appropriate corrections that enhance the manuscript.  I still do not understand how they used 400 ng of RNA template for cDNA with the final RNA concentrations measured using the spectrophotometer unless they used higher ones that were reported >50 ng/uL RNA (ie 7.8 uL 51 ng/uL RNA = 400 ng of RNA in a final volume of 20 uL. However, if I needed to reproduce this method to make cDNA, I would concentrate my RNA or resuspend RNA in less elution volume (50 uL) so no problem with the protocol.  

Correction: Figures 2-6. improve grammar. Eg. remove “a” from Line 273.

Author Response

The authors have addressed all my comments.  They have made appropriate corrections that enhance the manuscript.  I still do not understand how they used 400 ng of RNA template for cDNA with the final RNA concentrations measured using the spectrophotometer unless they used higher ones that were reported >50 ng/uL RNA (ie 7.8 uL 51 ng/uL RNA = 400 ng of RNA in a final volume of 20 uL. However, if I needed to reproduce this method to make cDNA, I would concentrate my RNA or resuspend RNA in less elution volume (50 uL) so no problem with the protocol. 

We add a sentence in the text (material and methods section). In most samples, the concentration and volume obtained during RNA extraction allowed the transcription of 400 ng of material as reported. Only in the few cases where the concentration was below 20 ng/ul was it necessary to concentrate the sample.

Correction: Figures 2-6. improve grammar. Eg. remove “a” from Line 273.

We do it.